# A Systematic Survey of Research Trends in Technology Usage for Parkinson’s Disease

**DOI:** 10.3390/s22155491

**Published:** 2022-07-23

**Authors:** Ranadeep Deb, Sizhe An, Ganapati Bhat, Holly Shill, Umit Y. Ogras

**Affiliations:** 1Analog Devices, Raleigh, NC 27603, USA; ranayatra@gmail.com; 2Department of Electrical and Computer Engineering, University of Wisconsin-Madison, Madison, WI 53705, USA; sizhe.an@wisc.edu; 3School of Electrical Engineering & Computer Science, Washington State University, Pullman, WA 99164, USA; ganapati.bhat@wsu.edu; 4Lonnie and Muhammad Ali Movement Disorder Center, Phoenix, AZ 85013, USA; holly.shill@dignityhealth.org

**Keywords:** Parkinson’s disease, wearable devices, digital health, biopotential devices, diagnosis, prognosis, taxonomy

## Abstract

Parkinson’s disease (PD) is a neurological disorder with complicated and disabling motor and non-motor symptoms. The complexity of PD pathology is amplified due to its dependency on patient diaries and the neurologist’s subjective assessment of clinical scales. A significant amount of recent research has explored new cost-effective and subjective assessment methods pertaining to PD symptoms to address this challenge. This article analyzes the application areas and use of mobile and wearable technology in PD research using the PRISMA methodology. Based on the published papers, we identify four significant fields of research: diagnosis, prognosis and monitoring, predicting response to treatment, and rehabilitation. Between January 2008 and December 2021, 31,718 articles were published in four databases: PubMed Central, Science Direct, IEEE Xplore, and MDPI. After removing unrelated articles, duplicate entries, non-English publications, and other articles that did not fulfill the selection criteria, we manually investigated 1559 articles in this review. Most of the articles (45%) were published during a recent four-year stretch (2018–2021), and 19% of the articles were published in 2021 alone. This trend reflects the research community’s growing interest in assessing PD with wearable devices, particularly in the last four years of the period under study. We conclude that there is a substantial and steady growth in the use of mobile technology in the PD contexts. We share our automated script and the detailed results with the public, making the review reproducible for future publications.

## 1. Introduction

Parkinson’s disease (PD) is a complex neurodegenerative disorder that affects patients’ and their caregivers’ overall quality of life (QoL). Approximately 60,000 individuals in the United States are diagnosed with PD each year, while more than 10 million people are living with PD worldwide [1,2]. Many significant motor signs accompany PD, such as tremors [3,4], rigidity [5], bradykinesia [6,7,8,9], hypokinesia [10], postural instability [11,12], and gait difficulties [13,14,15]. While clinical diagnosis is usually based on these motor symptoms, non-motor symptoms are also common and sometimes more disabling than motor symptoms. Non-motor signs of PD include cognitive impairment [16,17], dementia [18], depression [19], and emotional changes [19,20].

The current practice for assessing the motor and non-motor symptoms of PD patients is a neurological examination, during which a neurologist watches the patient perform specific tasks [21]. Neurologists assign scores to the tasks performed by the patient as described by the Unified Parkinson’s Disease Rating Scale (UPDRS) [22] or its updated version, the Movement Disorder Society-sponsored revision of the UPDRS (MDS-UPDRS) [23]. Another rating scale, the Hoehn and Yahr scale (HY) [24] assigns patients an overall score between 0 and 5 based on their clinical stage. Since clinical assessments rely on descriptions of symptoms’ progression recorded in patient diaries, their credibility is limited by the subjectivity and recall bias of the patient [25,26]. Imaging equipment, such as magnetic resonance imaging (MRI), single-photon emission computed tomography (SPECT), and positron emission tomography (PET), can assist the neurologist in making an objective and more accurate diagnosis [27]. However, high equipment costs factor into the expenses, increasing the total cost [28]. Medication alone can cost USD 2500 a year, while corrective surgery, such as deep brain stimulation, costs up to USD 100,000 per person [1].

Advances in sensing and processing technologies facilitate the long-term objective measurement of symptoms. Hence, new sensors, wearable devices, video capture/processing systems, and mobile technologies enable a wide range of monitoring, diagnosis, and rehabilitation applications [29,30,31,32,33,34,35,36,37,38,39]. Indeed, the number of publications that report using wearable and mobile technology for PD research increased by more than 20-fold from 2008 to 2021, as discussed in Section 4. Popular devices include inertial measurement units (IMUs), force and pressure plates, biopotential sensors, and optical motion capturing systems. This paper analyzes the trends of technology use in the PD context by analyzing articles published between 1 January 2008, and 31 December 2021. Starting with the keyword “Parkinson”, we performed an electronic database search in PubMed Central, Science Direct, IEEE Xplore, and MDPI to retrieve all articles related to PD. This search yielded 31,718 articles after we removed surveys and articles written in a language other than English. After completing the keyword-based automatic filtering of articles, we inspected the articles and classified them manually so that we could better understand their relevance to (1) application areas of the proposed solution, (2) symptoms measured for the application, and (3) devices and sensors used for measuring the symptoms. Then, we reviewed each article by performing four steps:Analyze the trends in use of technology for PD research. For example, we used the available data to identify the most popular application areas and the areas receiving increased attention (Section 2.2),Analyze the devices used in each application area (Section 2.3),Analyze the symptoms measured in each application area and identify the most commonly used devices for measuring the different traits (Section 4.1),Analyze the overall research trends in this field and how it progresses in the coming years (Section 4.2 and Section 4.3).

The potential benefits of this survey are threefold. First, it identifies the application areas and technology used in these areas. This information is beneficial to researchers new to PD and related diseases. Second, our trend analysis reveals the amount of attention received by each application area, symptom, and technology. This information can help researchers identify the gaps in the literature and determine the outstanding needs. Finally, our analysis shows which technologies have been applied to each application area and symptom. This information can guide researchers in choosing the best-known solutions and exploring new techniques that have not been tried yet.

The rest of this paper is organized as follows. Section 2 presents the related and background for our analysis. Section 3 describes our methodology, while Section 4 discusses the analysis results. Finally, Section 5 concludes the paper and discusses future directions.

## 2. Background

### 2.1. Related Work

Wearable and mobile devices can be carried by a user or be placed on any part of a subject’s body to collect relevant data using one or multiple embedded sensors. They either transmit the raw data via some communication interface (e.g., Bluetooth or Zigbee) to an external processing unit or process the data on a microcontroller in the devices themselves. An assortment of sensors, ranging from accelerometers, gyroscopes, and magnetometers to temperature, force, and pressure sensors, can be used individually or in various combinations for PD assessment [3,11,14,40,41,42,43,44,45].

Many recent studies have investigated the use of wearable sensors and other technologies to assess the symptoms presented by patients suffering from neurological disorders [46]. There is also a growing interest in performing unbiased analyses of the efficacy with which technology-based devices contribute to scientific research on health monitoring and clinical practices [47]. For example, Godinho et al. [47] reviewed 168 articles after searching the PubMed database and grouped them based on the device type. They classified the devices as recommended, suggested, or listed according to three criteria: (1) used in the assessment of PD, (2) examined in published studies by people other than the developers, and (3) successful clinimetric testing. The authors concluded that objective sensing technology is gaining attention in the study of Parkinson’s Disease but that the clinimetric properties and testing of the devices remain controversial. They surmised that the reviewed devices can objectively measure PD symptoms, such as postural control, bradykinesia, tremor, gait, and daily physical activity. Applications of wearable technology to assess PD symptoms have also expanded in recent years [48]. It has shown promise in PD diagnosis and management and other pathology [49,50,51,52]. An intriguing article on this topic by Daneault et al. [52] discusses how wearable and, more generally, mobile technologies could improve efforts to manage essential tremor, one of the most common movement disorders affecting millions of people worldwide. The study (1) proposes seven areas in which wearable and mobile technology can improve the clinical management of essential tremor, (2) reviews the current state of research in these areas, and (3) concludes that, by working together to address current knowledge gaps, various practitioners ranging from clinicians to engineers could leverage wearable and mobile technology everywhere and thus improve patients’ QoL.

Existing surveys focus on the technical contents of each proposed approach in the literature. They typically summarize the contributions, strengths, and weaknesses of particular methods [15,47,53]. In contrast, our analysis takes a holistic view to analyze macroscopic research trends. Hence, it can help researchers understand the current trends and identify the gaps in the literature. To our knowledge, this article presents the first systematic study that analyzes the research trends in wearable and mobile usage for PD. Our results summarize the use of wearable and mobile in terms of application areas, types of devices, and their recent trends.

### 2.2. Application Areas

Wearable and mobile technology can help medical practitioners assess the progression of PD patients by tracking their symptoms. These symptoms, referred to as cardinal PD features, are divided into motor symptoms and non-motor symptoms, as shown in Table 1. The motor symptoms are the movement impairments due to PD. They have traditionally been associated with tremor, rigidity, bradykinesia, and postural instability. Additionally, abnormalities in limb movements, such as hand rotation, finger tapping, and arm angle, have been regarded as motor symptoms. In contrast, non-motor symptoms include a large variety of cardinal features, such as sleep disturbance, cognitive activity, fatigue, dementia, and psychiatric impairments, as listed in the second column of Table 1. Finally, mixed symptoms, such as speech and swallowing difficulties, involve a combination of motor and non-motor activities. To show the prominence of these symptoms and justify their inclusion in our analysis, Figure 1 shows the number of articles that measure the different motor and non-motor symptoms. Gait abnormalities, including FoG, are the most common symptoms measured, followed by tremor. Movement problems, such as bradykinesia and dyskinesia, are also common symptoms in PD patients. Many research articles use modern measurement techniques to monitor these symptoms.

Our review and previous surveys of the PD literature [27,48] show that monitoring motor and non-motor symptoms can be helpful in four primary application areas in the PD research context: *diagnosis, prognosis and monitoring, predicting response to treatment, and rehabilitation*. The rest of this section reviews these areas.

#### 2.2.1. Diagnosis

PD diagnosis relies on neurologists’ clinical assessment of motor and non-motor symptoms [54]. To this end, neurologists observe patients while they perform specific tasks and assign them a score according to one of the standard scales: the previously mentioned UPDRS [22], its updated version, the Movement Disorder Society-sponsored revision of the UPDRS (MDS-UPDRS) [23], or the Hoehn and Yahr scale [24]. The clinical information derived from these rating scales is subjective. Hence, it leads to inter-rate variability and also intra-rate variability [27,48].

Moreover, as previously mentioned, the equipment used to supplement a clinical assessment tends to consist of expensive imaging tools, including MRI, SPECT, and PET. Because early and accurate diagnosis is essential, medical practitioners hope to augment current techniques with objective and cost-effective alternatives enabled by wearable and mobile technology. Therefore, recent research has examined the possibility of using wearable sensors and other portable technology to diagnose PD. These devices can provide objective PD-diagnosis measures that help standardize assessments [55,56,57]. Many researchers have also been able to use wearable sensors and other devices to distinguish PD patients from healthy controls in lab experiments [55,56,58].

In summary, wearable and mobile technology is used in the following sub-categories of PD diagnosis:Early diagnosis of patients with PD;Detecting PD symptoms in people with untreated PD;Distinguishing PD patients from either healthy controls or patients with non-PD neurological disorders;Distinguishing PD-related symptoms from similar symptoms not caused by PD (e.g., distinguishing PD tremors from essential tremors).

#### 2.2.2. Prognosis and Monitoring the Severity of Symptoms

Predicting the patient’s future condition (prognosis) and assessing the severity of the disease, including current symptoms, are two activities that depend primarily on clinicians’ judgments and patients’ diary- or memory-based feedback. Clinicians’ judgments are subjective [27,48], while patients’ diary entries and memories are limited by compliance and recall bias [25,59,60]. Because clinical approaches that rely on these sources of information may not be completely reliable, the objective remote monitoring of PD symptoms holds immense promise for assessing the disease progression, evaluating symptom severity, and monitoring PD patients in unsupervised environments. To address these issues, recent work on PD prognosis and monitoring has focused on the following areas:Home-based or remote monitoring of PD patients;Evaluating the PD progression in a diagnosed patient;Evaluating the severity of PD symptoms for a diagnosed patient.

#### 2.2.3. Predicting Response to Treatment

To measure the efficacy of a treatment, clinicians rely on patients’ diary entries and unrecorded memories, both of which, as noted, can be subjective and unreliable. This problem has motivated researchers to measure the effects of treatments, including the effectiveness—and the side effects—with which medications suppress PD symptoms. To this end, researchers studying the ability of practitioners to accurately predict a patient’s response to treatment have been addressing the following issues:Measuring the effects of such treatments as deep brain stimulation on the suppression of patients’ symptoms over time;Measuring the intended effects of medications on patient symptoms;Measuring the unintended effects (i.e., the side effects) of medications on patients (e.g., levodopa-induced dyskinesia).

#### 2.2.4. Rehabilitation

Physiotherapy and other rehabilitation techniques are among the most common treatments for PD and, indeed, for movement disorders in general. As with medications, it is crucial to assess the efficacy of rehabilitation techniques. Additionally, it has been observed that cues and feedback are beneficial in assisting a PD patient. A rehabilitation system can provide auditory, visual, or haptic cues to facilitate a patient’s movement. Such a system can be explicitly used for gait training and assessing limb movements. Vibration-based actuators and audio feedback help patients suffering from rigidity, FoG, tremors, and other symptoms. These approaches can also help patients break out of a freeze and even suppress symptoms. Mobile technologies that target therapy and rehabilitation can be divided into the following sub-categories:Audio, visual, or haptic cue for gait or movement training;Sensory feedback to suppress a symptom such as FoG or tremor.

### 2.3. Technology in Parkinson’s Disease Research

Our review and other surveys of the PD literature [27,48,61,62] reveal that researchers use a number of device types and technologies in the application areas presented in the previous section. Using the current literature, we classify these devices into the following eight categories based on the device form factor and sensing modalities.

#### 2.3.1. Wearable Devices

Many recent PD-oriented approaches focus on wearable devices for health monitoring since they help medical practitioners record patients’ activities and symptoms. Wearable devices are ideal monitoring technologies for two main reasons: they are not limited to a specific location and can be easily integrated into patients’ clothes [63]. The most commonly used wearable devices contain sensors, such as IMUs. The outputs of these sensors can facilitate analyses of body movement, gait, and symptoms, such as tremors. Our study covers the following wearable technologies:IMUs with integrated accelerometers, gyroscopes, and magnetometer sensors;Insole force- or pressure-based sensors that can measure the ground reaction force (GRF);Wearable devices that use sensors such as EEG and EMG to measure neural responses and muscle activities;Clothing-integrated sensors, such as strain or accelerometer sensors, that measure hand tremors;Other wearable devices, including smart glasses and smart hats, that can record patients’ emotions and other specific parameters.

#### 2.3.2. Biopotential Devices

These devices can measure the electrical signals (biopotentials) generated by our body’s physiological processes. This category includes electroencephalogram, electrocardiogram, magnetoencephalogram, and electrooculogram devices. They can be either standalone devices, such as a 16-lead EEG system, or wearable devices with a single-lead EMG sensor. When a biopotential device is integrated into a wearable device, the resulting device falls under both the wearable and biopotential-device categories.

#### 2.3.3. Cueing Devices

Devices that give feedback or cues to patients to rectify their walk or assist with other movements fall under this category. Headphones or speakers can deliver auditory cues, and visual cues can appear on a screen, smart glass, or even in a virtual-reality environment. Vibration sensors and electrical stimulators also can give sensory feedback to patients.

#### 2.3.4. Optical Motion Tracker

These devices track user motion with either radio frequency or optical signals. For example, motion-capturing systems such as Microsoft Kinect [64] and Vicon 3D [65] use structured light for analysis. Similarly, radar-based systems use radio-frequency signals to monitor the motion of subjects [32,37]. A system of multiple IMUs, even camera-based 3D setups, can also capture a patient’s movement.

#### 2.3.5. Audio Recording

Many PD patients experience issues with swallowing or speech, as shown in Table 1. Audio recording devices can analyze and monitor symptoms related to swallowing and speech. For instance, microphones and smartphones record speech tasks or patients’ voices for the subsequent diagnostic analysis [58,66].

#### 2.3.6. Video Recording

Video-recording devices are helpful for PD monitoring because gait and motor issues are among the most common PD symptoms. For instance, video cameras record patients’ movements at home or during laboratory experiments, and practitioners can use the recordings to spot symptoms or to corroborate or disprove predictions based on other devices [6,20,67].

#### 2.3.7. Force/Pressure

Force or pressure plates measure the force exerted by the patients’ feet while walking. Therefore, they help measure the gait quality of the patient. Force or pressure sensors are also integrated into gait mats to measure the gait quality of a patient.

#### 2.3.8. Smartphone

Researchers have used accelerometers, gyroscopes, magnetometers, GPS, and other sensors on smartphones for various analytical objectives. Smartphone applications can also record such matters as patients’ emotional displays and drug dosages. Additionally, smartphone screens can record handwriting, microphones can record speech, and cameras can record movement.

#### 2.3.9. Other

Other devices have also helped practitioners assess PD patients. We classify these devices that do not fall into the above categories as “other.” For example, digitized tablets can serve as smart screens for handwriting assessment. PD patients can write on these screens during spiralography exams. Likewise, smart pens can record hand movements during writing. Solutions based on virtual or augmented reality have also started becoming popular in PD applications.

## 3. Methodology for Article Selection and Classification

### 3.1. Search Methodology

The first step in our analysis was obtaining the original PD-related articles published between 1 January 2008 and 31 December 2021, as shown in Figure 2. To this end, we performed an electronic database search of articles published during this period in the PubMed Central, Science Direct, IEEE Xplore, and MDPI databases. We chose these databases so that our search process would cover both medical and engineering journals [15].

We performed our systematic searches by following the PRISMA guidelines [68]. To make the search as broad as possible, the initial search query consisted of just one keyword, “Parkinson”, in the title, abstract, and keyword (TAK). Table 2 lists our search queries and the corresponding number of hits in each database. From the queries, we obtained a total of 31,718 articles, of which 14,898 were from PubMed Central, 12,216 were from Science Direct, 2564 were from IEEE Xplore, and 2040 articles were from MDPI.

**Exclusion criteria during the initial search:** From this pool of articles, we removed the ones that were not written in English or that were review articles themselves. At this point, we retrieved specific information from each article: namely, its title, authors, publication title, year of publication (YOP), keywords, abstract, and DOI. Then, as discussed in the next section, we put the resulting data through the filtering step to exclude articles that fell outside the scope of our research.

### 3.2. Filtering Methodology

We developed an automated script in Python to filter the information collected from the four databases. Following the PICO strategy [69], our script uses four keyword blocks adapted from [70] to implement the selection and exclusion criteria. The first keyword block, which has 10 PD-related keywords, specifies the inclusion criteria in terms of the PD terminology and symptoms. After selecting the articles with at least one of these keywords in TAK, the second block, which has 7 keywords, was used to exclude the studies conducted on non-human subjects, as shown by the first two blocks in Figure 3. Next, we filtered the articles according to the type of device discussed with the next set of keyword blocks. Specifically, the third keyword block has 66 terms for 66 technology devices commonly used for PD assessment. This block is illustrated by the third block in Figure 3, while the complete list of keywords is provided in Appendix C. Finally, we used the fourth keyword block, which consists of 11 terms, to exclude 11 technologies that are currently unsuitable for use outside the four walls of an actual clinic. Five such technologies are MRIs, deep brain simulation, PET, neuroimaging, and SPECT. A complete list of all keywords is provided in Appendix C.

The automated filtering process excluded 28,284 articles from the complete list of 31,718 articles. Specifically, the first keyword block included 9572 articles relevant to the symptoms under review herein. From these 9572 articles, the second keyword block excluded 4982 articles because they used non-human subjects. This step thus left 4590 articles for further filtering. Similarly, the third and fourth keyword blocks excluded 1156 articles, as they involved studies about technologies that fell outside the scope of our current review. Appendix C lists all of the keywords used in these keyword-block steps. Finally, the script removed duplicate entries, leaving 3319 articles. The Python code used for filtering and removing the duplicate articles will facilitate future studies, particularly those addressing relevant articles published after 2021.

### 3.3. Classification Methodology

We manually analyzed the final set of filtered articles to classify them according to application area and technology used. We first read the title, abstract, and keyword (i.e., TAK) sections to understand the main application area of each article. Specifically, we chose one of the four application areas described in Section 2.2 and identified the symptoms (Table 1) analyzed by the articles. When we were unable to extract the application area or the addressed symptoms from our TAK review of an article, we read the entire article to accurately classify it in relation to these two matters. Using a similar procedure, we categorized the devices and the other technology addressed in each article.

As part of the manual classification, we also exclude articles that fell out of the scope of our review. For example, despite falling outside this scope, some articles use specific terms, such as “sensor” and “acceleration”, which initially led them to be included in our review, hence the need for manual classification. Similarly, the script accidentally included some articles because they featured the term “ECG”, which in these cases, was related not to PD but “Wolff–Parkinson–White (WPW) syndrome”. We manually excluded such articles from the final list of classified items. Overall, we identified that 1559 of the 31,718 articles were relevant and classified according to their technology and application areas.

## 4. Results and Discussion

This section analyzes the research trends in the selected articles published between 2008 and 2021. Before describing the details of the research trends, we first present the publication trends during this period in Figure 4. Overall, the total number of articles published yearly increased steadily, with an accelerated rate in the last four years (2018 to 2021). Notably, most of the articles (45%) were published in the last four years, while an impressive 19% of the articles were published in 2021 alone. This trend shows the growing interest in finding alternative objective ways of assessing Parkinson’s Disease, particularly in recent years. In the following, we first analyze the number of articles published in each application area. Then, we focus on the various PD symptoms studied in these articles. Finally, we describe the various wearable and mobile technologies used in these studies. Our analysis in this section can help researchers understand the changes in PD research over the last few years and identify future directions in this area.

### 4.1. Application Areas

Wearable technology has varying levels of utility in each application area of PD, as the pie chart in Figure 5 shows. Of the 1559 articles evaluated, 719 (46%) focus on PD diagnosis (including diagnostic assistance) and 479 (31%) focus on PD prognosis and monitoring. Furthermore, 236 articles (15%) address patient rehabilitation and patient QoL, and 125 articles (8%) analyze the effects of medicine and other treatments on patient symptoms. Figure 5 shows that the most frequently targeted application areas during 2008–2021 are diagnosis and prognosis/monitoring disease progression. The following sections detail the trends in each application area.

#### 4.1.1. Diagnosis

Being well aware that a correct diagnosis is of prime importance, many researchers have proposed objective and inexpensive technology-based PD diagnostic techniques. Figure 6a shows that the number of articles on PD diagnosis increased steeply and surpassed 150 in the last two years. A more detailed investigation indicates that studies analyzing both motor and non-motor symptoms contributed to this increase. For example, Raethjen et al. [71] and Zhang et al. [72] used EEG and EMG data to characterize tremor in PD patients. Similarly, gait analysis using smartphone accelerometers successfully distinguished PD patients with gait disorder from healthy controls [55]. Muscle-activity information obtained from EMG sensors has also proved helpful in PD diagnosis. For example, Meigal et al. [56] employed surface EMG (sEMG) to distinguish between PD subjects and healthy controls and analyzed the severity of the former subjects’ symptoms. Studies focusing on non-motor symptoms analyzed sleep patterns and speech disorders as PD biomarkers. With EMG data recorded from the chin of a sleeping PD patient, researchers conducted a comparison between the rapid eye movement (REM) of PD patients with REM sleep behavior disorder (RBD) and PD patients without RBD [57]. Similarly, Campos-Roca et al. extracted various acoustic features from an acoustic data set of 40 healthy control and 40 PD patients [58]. In another study, Tsanas et al. used four parsimonious subsets of 132 dysphonia features from an existing data set of 263 samples drawn from 43 subjects [66]. The authors showed that the classification with the new dysphonia features reached almost 99% accuracy. These studies show that recent technological advances combined with novel algorithms can help clinicians develop objective measures for PD diagnosis. Table A4 in Appendix B provides a comprehensive list of the articles focusing on diagnosis.

#### 4.1.2. Prognosis and Monitoring Disease Progression

Wearable and mobile technologies are well suited for accurate home-based prognosis and monitoring because they can continuously record patients’ movements and symptoms. Indeed, 479 (31%) studies included in our analysis deal with PD prognosis and monitoring, as shown in Figure 5. The most commonly analyzed problems concern the tracking of PD patients’ symptoms, assisted living for PD patients, the evaluation of PD-symptom severity, and disease progression in general. Figure 6b presents the growth trends in these prognosis-and-monitoring articles during the 2008–2021 period. The growth picked up a considerable pace after 2014 and approached 75 papers per year. This growth is driven mainly by improvements in wearable technologies. Table A3 in Appendix B provides a comprehensive list of articles about prognosis and monitoring.

The articles on prognosis-and-monitoring applications have focused on enabling clinical tests outside the clinic. In many cases, these applications are helpful for both pre-prognosis monitoring and post-prognosis monitoring. For example, the “Timed Up and Go” test (TUG) is a commonly used clinical test to evaluate balance and mobility. Salarian et al. proposed an instrumented TUG called iTUG using portable inertial sensors [73]. PD monitoring is an essential contributor to the development of accurate prognoses and post-prognosis assessments. Thus, many studies focus on the monitoring of gait parameters. For instance, Zwartjes et al. [6] presented an ambulatory monitoring system that provides a complete motor assessment of PD patients by simultaneously analyzing their motor activities and the severity of several symptoms, including tremor, bradykinesia, and hypokinesia. Another study in 2016 proposed that smartwatches can help quantify tremors in PD patients [4]. Similarly, Bächlin et al. [74] proposed a wearable assistant to detect FoG events during ambulatory movements. Researchers have recently focused on non-motor symptoms, such as emotions and fatigue, to assess PD progression. One study recorded the facial expressions of 40 PD patients to investigate the relationship between reduced facial expressiveness and altered emotion recognition in PD [20]. Overall, these studies demonstrate the growing potential of technology to enable increasingly precise and accurate monitoring of PD patients. Hence, the growth trend evident in Figure 6b is likely to continue in the coming years.

#### 4.1.3. Predicting Response to Treatment

About 8% of 1559 studies in our analysis measured the efficacy of PD treatment in patients. Figure 6c illustrates trends in the number of articles published annually between 2008 and 2021 regarding clinicians’ ability to predict patients’ responses to therapy. The growth in this application area was steady throughout the period, except for an outlier jump in 2019. In general, this steady growth suggests that the focus on using wearable and mobile technology to predict PD treatment responses did not experience a massive shift during the period. Table A1 shows a summary of the papers that focus on predicting response to treatment, while the rest of this section highlights the notable therapies for PD and technology for predicting the response.

Wearable and mobile technology may help predict patients’ responses to several powerful, frequently used PD treatments. Levodopa medication is the most popular treatment for motor and non-motor PD symptoms. It helps alleviate important symptoms, such as bradykinesia, rigidity, and tremors. Another conventional treatment methodology is deep brain stimulation of the subthalamic nucleus. The stimulation helps alleviate motor symptoms and, thus, reduces the dosage of dopaminergic medication. Measuring the efficacy of such treatments in PD patients is essential since the mechanism through which the treatments improve cognitive or motor operations is not well understood. Recent studies have proposed EEG, EMG, and other biopotential methods to measure the effects of levodopa medication on PD patients [7,75,76]. In contrast, Rigas et al. [77] and Pelicioni et al. [78] proposed new symptom-assessment methods for PD patients using wearable sensors. Other research articles have proposed wearable-sensor approaches to measuring the effectiveness of deep brain stimulation in PD patients [10,79].

In summary, most of the articles whose focus is on predicting response to treatment have analyzed the severity of PD symptoms before, during, and after treatment. Some studies have also focused on understanding how a given medication alleviates PD symptoms. Further research in this domain will help health professionals fine-tune treatment for individual patients according to their individual responses to specific treatments.

#### 4.1.4. Rehabilitation

Developing an efficient rehabilitation plan for PD patients is crucial to managing their symptoms. Rehabilitation techniques may use assistive cues to help the patients with daily movement or activities. Between 2008 and 2021, 236 articles were published regarding rehabilitation techniques. Significant research also focused on developing methods to alleviate motor symptoms, such as FoG and tremors, by using auditory and haptic (e.g., vibratory) cues. Table A2 lists the articles related to rehabilitation. Below, we summarize recent approaches to rehabilitation.

The decline in motor and cognitive functionalities can significantly increase the risk of falling and the incidence of FoG while reducing QoL [15,80]. To counter the decline in motor functions, clinicians often employ the rehabilitation technique known as stepping-in-place training [81]. Sensory cuing can facilitate stepping-in-place for PD patients. Audio recordings of specific tasks can effectively reduce gait variability in PD patients but not reduce FoG. Young et al. studied the efficacy of such auditory cues in PD patients with FoG [82]. Similarly, haptic cues have been used in various studies whose objective is to unfreeze a PD patient’s gait during an episode of FoG [83,84]. Vidya et al. [85] combined pulse-width modulation with a coin-type vibration motor on patients’ wrists and a micro-controller to generate random vibration patterns to help PD patients cope with hand tremors. In summary, the interest on improving rehabilitation strategies for PD patients steadily increased between 2008 and 2021, as shown in Figure 6d. The growth was linear, with the number of papers increasing from about two in 2008 to around 25 in 2021.

### 4.2. Trends in Symptoms Measured by PD Research Papers

During the period under examination, much research explored how clinicians should assess motor symptoms since they are often the most visible symptoms in PD patients. Inertial data collected by wearable devices are broadly used to monitor gait parameters, tremors, motor activities, FoG events, bradykinesia, and dyskinesia (the on-off stages). Force and pressure sensors placed under the shoe or in an insole can measure the ground reaction force, a popular parameter for analyzing gait. EMG sensors can monitor the muscular response of a person. More sophisticated instruments, ranging from digitized tablets to smartpens, help clinicians analyze the hand movement and pressure experienced by PD patients engaged in hand-writing. The heat-map in Figure 7 shows the number of times a specific type of motor symptom was the basis for an article’s proposed solution for a particular PD application area. Gait abnormality is the most frequently cited motor symptom in PD assessment across all application areas except for the “Predicting Response to Treatment” area. In that area, bradykinesia and dyskinesia are the motor symptoms that articles most commonly monitor. These articles primarily monitored the on-off stages and evaluated muscle activities. Tremor and FoG are common motor symptoms in PD patients and biomarkers for objective assessments of PD. Analyzing the balance and posture of PD patients is a common strategy used in prognosis, monitoring, and rehabilitation.

Figure 8 presents the trends in the research community’s focus on gait, tremor, and other PD symptoms during the 2008–2021 period. Following 2012, more articles focused on gait than on tremor, even though the annual number of articles addressing one or the other of these symptoms kept growing. With the development and popularity of new wearables and advanced IMUs, measuring gait parameters became easier.

By 2021, many researchers focused on non-motor symptoms such as cognitive impairment, dementia, and depression. Since these symptoms can be more disabling for a patient, an objective assessment is required. Analyzing neural response measurements is a common strategy used to improve their diagnosis, monitoring, and analysis of responses to treatment, as shown in Figure 9. Analyzing the cognitive activity of PD patients is also a strategy used in various PD application areas. However, it is evident from the Figure 8 heat-map that the amount of research focusing on severe and disabling symptoms, such as dementia, depression, and fatigue, is far smaller than the amount of research focusing on motor symptoms.

### 4.3. Trends in Device Usage from 2008 to 2021

This section evaluates the types of technology employed by the selected articles under review. Of the 1559 reviewed articles, 40% used wearable devices, 18% used biopotential devices, 13% used audio-recording devices, 5% used smartphones, and 3% examined auditory, haptic, or visual cuing devices, as summarized in Figure 10. This figure shows that wearable and biopotential devices are the most common technologies.

Biopotential devices applying EEG, ECG, EMG, EOG, and similar technologies to PD-stage assessment were common topics. Several articles examine EEG recordings for the measurement of patients’ neural activity, which is a popular PD biomarker [71,86,87,88]. Other articles examine EEG data for clinical assessments of non-motor symptoms, such as sleep, dementia, and cognitive activity, and of mixed symptoms, such as saccades [16,89,90,91,92,93]. Similarly, EMG recordings are used for muscle activity, which is a factor instrumental in PD diagnoses and prognoses [94,95,96,97,98]. Many articles discuss how ECG and EOG technology can enhance clinicians’ study of heart rates and optical movements for assessing PD in patients [99,100,101]. Figure 11a,b illustrate the growth in the number of articles devoted to wearables or biopotentials. Figure 11a shows that the number of articles examining biopotential devices for the evaluation of PD symptoms grew at a constant rate during the 2008–2021 period. With ongoing advances in portable biopotential devices and wearable devices featuring built-in biopotential sensors, it is reasonable to anticipate that PD research using this technology will continue to grow.

Section 2.3.1 elaborates the different devices categorized as “Wearable”. Any wireless device placed on any part of a subject’s body to collect relevant data can be called a wearable device. Different types of sensors, such as the accelerometer, gyroscope, magnetometer, temperature, force, and pressure, have been used individually or together for PD assessment [3,11,14,38,39,40,41,42,43,44,45]. Some articles examined the incorporation of EMG and other biopotential sensors into wearable devices to collect data about PD patients’ muscle activity [72,76,102,103,104,105]. Several other articles evaluated sensors, including insole force (i.e., pressure) sensors, which generate data on the vertical ground reaction force generated when a subject is walking: using these data, clinicians can assess the subject’s gait, balance, or posture, all of which are critical factors for PD patients [106,107,108,109,110,111,112]. Figure 11b shows that the number of articles using wearable devices increased significantly during the period under examination.

**Device Usage Trend:** The heat map in Figure 12 tracks the number of articles using at least one of the cited devices during the 2008–2021 period. We see that research on all the devices continued to grow. The growth, which was significantly pronounced for wearable and biopotential devices, can be attributed to advances in sensor technology. As with wearable and biopotential devices, the last several years of the period under review witnessed a growing number of articles using motion-capturing systems, audio-recording devices, and smartphones).

**Device Usage by Application Area:**Figure 13 shows the distribution of device categories across the four primary application areas. Almost all device categories, except cuing, were used predominantly in diagnosis, prognosis, and monitoring. This trend was expected because most articles in our review focused on one or the other of these two application areas. We also discovered that wearable devices were primarily used in the prognosis-and-monitoring application area because these devices can effectively monitor patients in a free-living environment. For similar reasons, motion tracking and video recording were extensively examined with prognosis objectives. Biopotential devices were one of the most heavily studied devices in predicting response to therapy. In the articles, audio-recording devices were prominently studied regarding their potential to aid in diagnosis but were seldom studied to help predict response to treatment. Interestingly, not a single article examined smartphones for their potential application in predicting response to treatment.

We also observe that cuing devices are used almost exclusively for rehabilitation. Specifically, about 90% of the articles focusing on cuing systems did so in the context of rehabilitation. Researchers used active cuing such as vibration and auditory feedback to suppress symptoms such as FoG and tremor. Various scientific studies used visual cues to facilitate walking among PD patients [13,113,114]. Wireless headphones were also studied to give patients auditory feedback that would improve their motor activity, gait training, and balance training, all of which are forms of rehabilitation [13,115,116,117]. Moreover, several articles centered on auditory and vibratory cues for their possible contribution to the FoG- or tremor-related rehabilitation applications [74,83,85,118,119,120,121,122].

## 5. Conclusions, Gaps in Literature, and Future Directions

This paper presents a comprehensive overview of the technological solutions currently being used for their potential contributions to the objective assessment of PD. After reviewing 1559 relevant articles published between 2008 and 2021, we identified four primary application areas addressed by researchers as diagnosis, prognosis and monitoring, predicting response to treatment, and rehabilitation. Our analysis also showed the most commonly measured symptoms, including gait, tremor, limb movements, and FoG. Moreover, our analysis revealed the most commonly used technologies, led by wearable devices, biopotential sensors, and audio recording. The publication trends show that the number of studies in each application area and technology use has increased steadily, with a larger rate in recent years. We also note that novel contributions made by the papers included in the survey are important to understanding the use of technology in PD. For instance, analyzing the contributions of each study can help us understand if any particular symptoms are more challenging to monitor than other symptoms. At the same time, judging the contributions of different approaches is difficult since it is subjective. Therefore, we did not include such an analysis in this paper and leave it for future work.

We concluded that the scientific community undertaking these studies has generally sought to use unobtrusive systems to monitor PD progression, beginning in its nascent stage. Diagnosis has received the most significant attention due to its intuitive importance, although it is arguably one of the two most challenging problems along with treatment of PD. Therefore, a notable gap in the literature is the insufficient attention to prognosis, monitoring, and response to treatment. Progress in these fields can both directly benefit PD patients and pave the way for established diagnostic assistance. Regarding technology, most studies employ a single modality, such as wearable inertial sensors. This trend can be attributed to the nontrivial expertise required for using a specific technology as well as integrating multiple sensing modalities. However, significant capability and quality improvements are possible with multi-modal techniques that combine different technologies, such as wearable devices with audio and video recording. Few studies take advantage of multiple modalities, but a systematic combination of carefully fused devices remains a significant gap in the literature. We hope that this survey will catalyze research in using technology for PD monitoring and treatment to close the gaps. Our future work includes developing algorithms for the automatic classification of papers to minimize the need for manual inspection of papers. This automation will also enable new surveys targeting different movements and neurological disorders, such as Alzheimer’s.

## Figures and Tables

**Figure 1 sensors-22-05491-f001:**
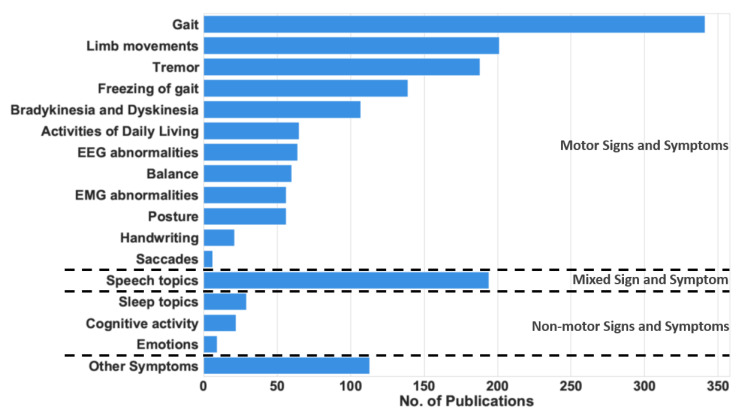
The number of research publications between 2008 and 2021 that measure each sign and symptom.

**Figure 2 sensors-22-05491-f002:**
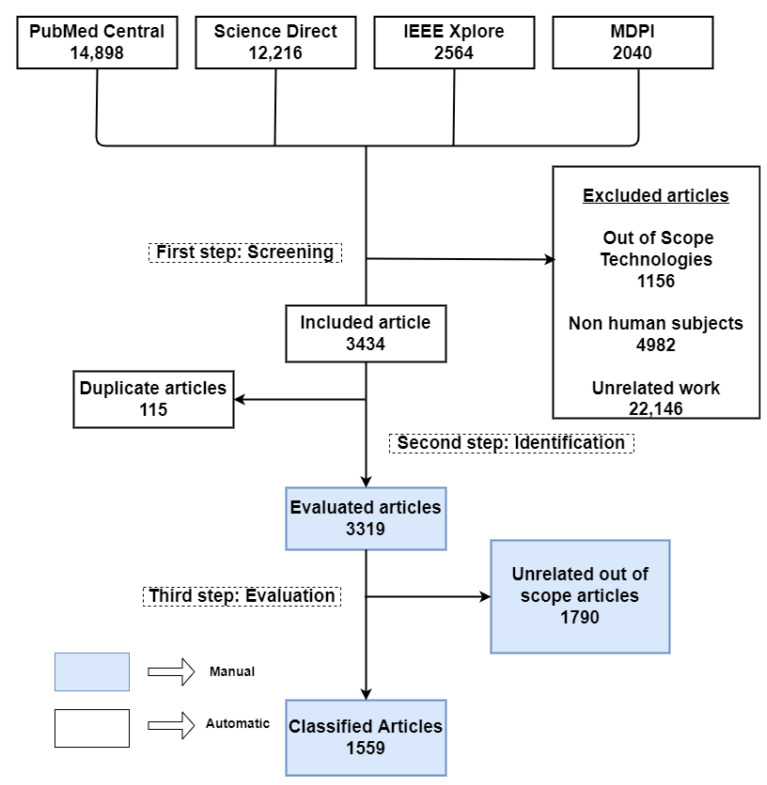
Flow diagram of our systematic review process for PD-assessment research publications (2008–2021).

**Figure 3 sensors-22-05491-f003:**
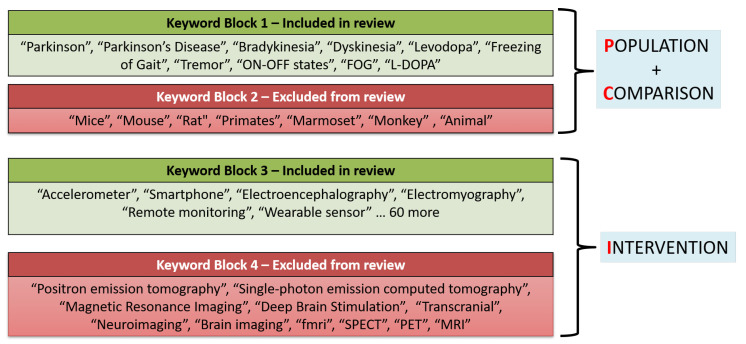
Keyword blocks constructed according to the PICO strategy to determine the relevance of an article to this review.

**Figure 4 sensors-22-05491-f004:**
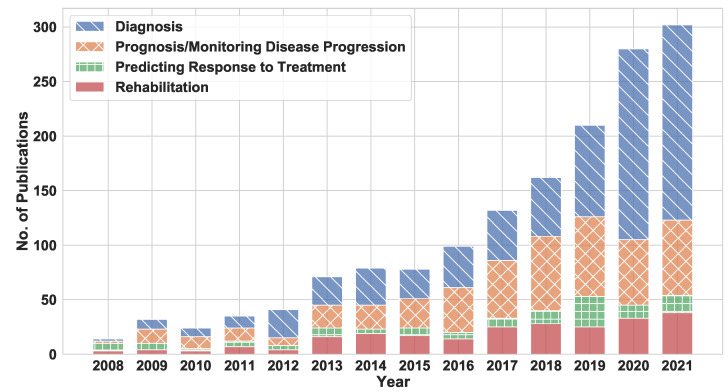
The yearly publication trends between 2008 and 2021 in each application area.

**Figure 5 sensors-22-05491-f005:**
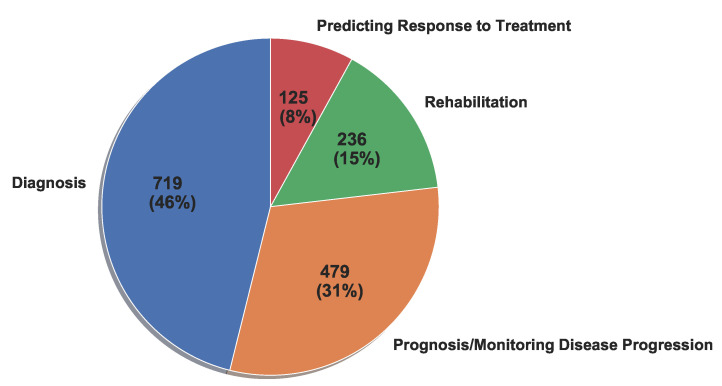
Percentage of publications (2008–2021) by application area.

**Figure 6 sensors-22-05491-f006:**
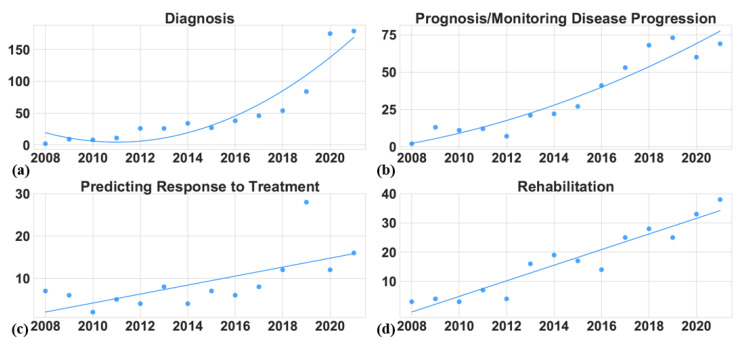
The number of research publications (2008–2021) divided by PD application area: (**a**) publications about diagnosis, (**b**) publications about prognosis and monitoring, (**c**) publications about predicting patient responses to treatments, and (**d**) publications about rehabilitation.

**Figure 7 sensors-22-05491-f007:**
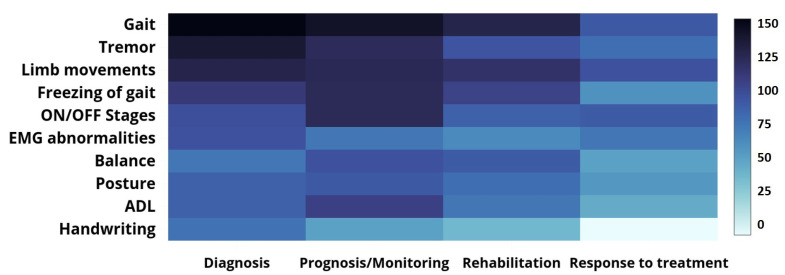
The heat-map presenting the number of research publications (2008–2021) that address the measuring of motor PD symptoms (the darker the color, the higher the number).

**Figure 8 sensors-22-05491-f008:**
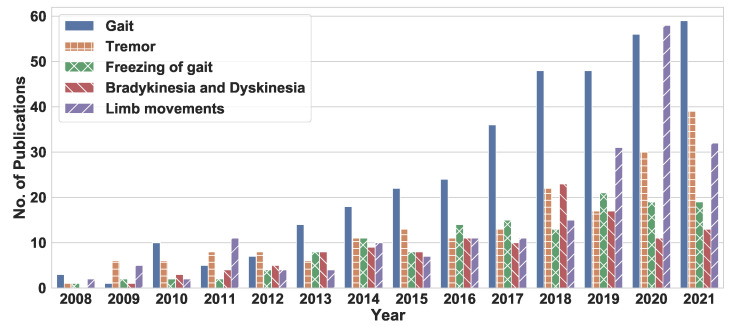
The number of research publications (2008–2021) examining the application of new technologies to specific types of PD symptoms (e.g., FoG, and tremor).

**Figure 9 sensors-22-05491-f009:**
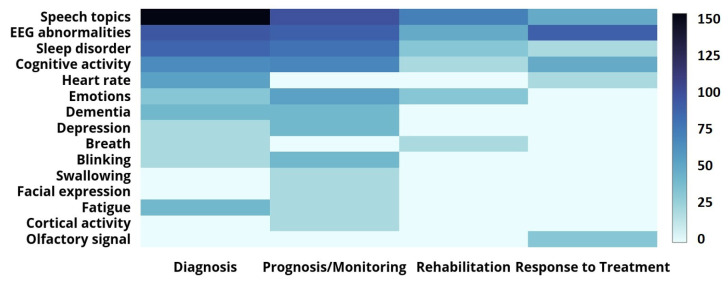
The heat-map presenting the number of research publications (2008–2021) that address the measuring of non-motor PD symptoms (the darker the color, the higher the number).

**Figure 10 sensors-22-05491-f010:**
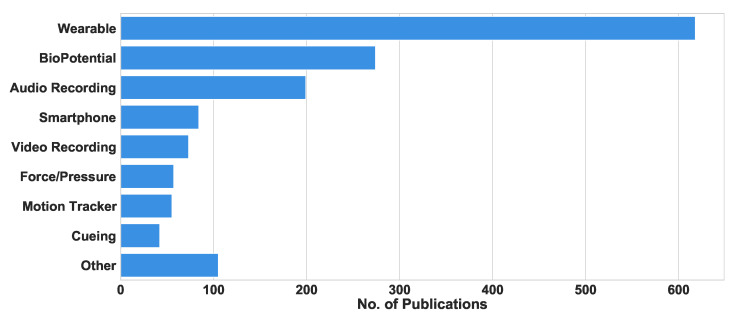
Number of research publications (2008–2021) examining the application of specific types of new technologies to PD areas.

**Figure 11 sensors-22-05491-f011:**
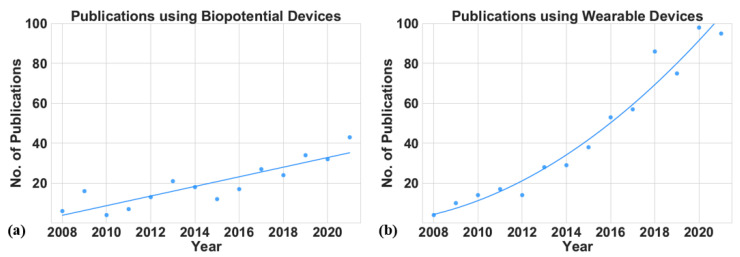
The number of research publications between 2008 and 2021. (**a**) Studies that use biopotential devices for PD assessment, (**b**) studies that use wearable devices for PD assessment. The solid lines shows the trend of the publications in the last 14 years.

**Figure 12 sensors-22-05491-f012:**
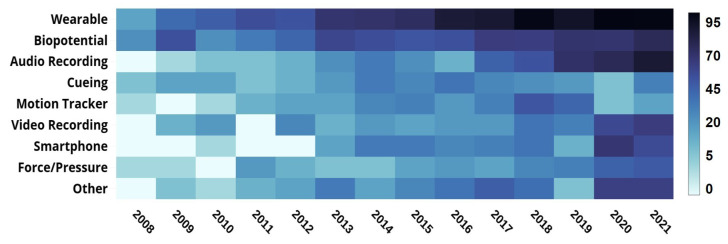
The heat-map presenting the number of research publications (2008–2021) that address PD assessment using modern technology (the darker the color, the higher the number.

**Figure 13 sensors-22-05491-f013:**
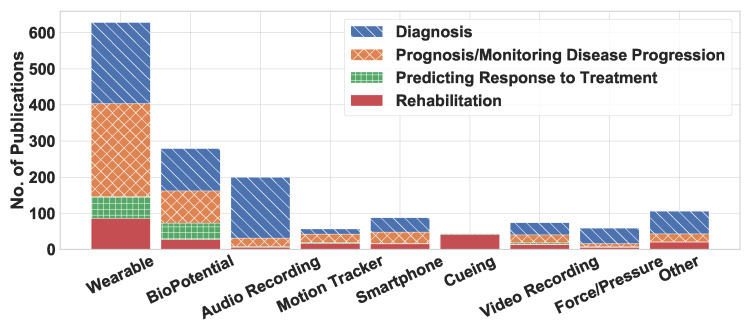
The distribution of articles published between 2008 and 2021 related to the assessment of Parkinson’s disease using wearable and mobile technology.

**Table 1 sensors-22-05491-t001:** The signs and symptoms assessed by technology in the current literature.

Motor Signs and Symptoms	Non-Motor Signs and Symptoms	Mixed Signs and Symptoms
Gait	EEG abnormalities	
Limb movements	Cognitive activity	
EMG abnormalities	Depression	
FoG	Dementia	
Tremor	Heart rate	Speech topics
Activities of Daily Living (ADL)	Emotions	Swallowing
Bradykinesia and Dyskinesia	Fatigue	
Posture	Sleep topics	
Balance	Blinking	
Nocturnal Hypokinesia	Facial expression	
Handwriting	Breath	
Saccades	Cortical activity	

**Table 2 sensors-22-05491-t002:** Search queries used for each database.

Database	Query	Years	Hits
Pubmed Central	parkinson [Body-Key Terms] OR parkinson [Abstract] OR parkinson [Title]	2008–2021	14,898
Science Direct	“parkinson” in Abstract OR Keyword OR Title	2008–2021	12,216
IEEE Xplore	“parkinson” in Abstract OR Keyword OR Title	2008–2021	2564
MDPI	“parkinson” in Abstract OR Keyword OR Title	2008–2021	2040

## Data Availability

The details of all the evaluated papers are listed in the publicly shared table made available at https://github.com/SizheAn/Parkinson-s-Disease-Survey (accessed on 21 July 2022).

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
