# Peer review of "A Systematic Survey of Research Trends in Technology Usage for Parkinson’s Disease"

_sensors, 2022, doi:10.3390/s22155491_

Round 1

Reviewer 1 Report

This work entitled "A Systematic Survey of Research Trends in Technology Usage for Parkinson’s Disease" aimed to present a systematic review of mobile-technology research in the PD context by analyzing articles published between January 1, 2008, and December 31, 2019.

Some observations:

  • Line 27 [1?] it's a typos erroe i think, please check the whole manuscript for this type of errors.
  • Line 46 [?]
  • Table and figures are clear and easy readable. The information are detailed described.
  • The work correctly follow the PRISMA guidelines, but i can't find a paragraph about Risk of Bias, this is an important aspect in a sistematic review, if for some reason in this case is not necessary authors have to explain why.
  • Reference list is lacking of some work of Sawacha et al. which could help authors to improve their intriduction section.

Reviewer 2 Report

Major comments:

The paper is difficult to follow. It would be great to re-organize the paper with concise information on background, method, results and discussion, and conclusion sessions.

The introduction session is lengthy, but why such a survey is needed, and the study’s objectives were not clearly presented.

For the application areas, were these areas summarized based on the survey results? If not, why were those areas used? Figure 1 should be the results, right? The four areas: diagnosis, prognosis, predicting response to treatment, and rehabilitation should be clearly defined. If a study is to predict response to treatment, should it be considered as rehabilitation as well?

For the technology in Parkinson’s Disease Research session, were those the survey results? Why were they here before the method of the survey?

Figure 3 of the keyword blocks was not the correct usage of the PICO strategy. The comparison is for whether a control group or a comparison group was included in a study. Many of the articles included in this survey were not interventional studies which are not applicable to following the PICO strategy.

The automated filtering is of interest and considered innovative but not clear how it worked.

From the results, what readers can learn? What will be the contribution of the results to PD individuals, their family members, clinicians, technology developers, and researchers? It would be great to list key takeaways for different stakeholders.

The inconsistency among keywords, such as “mobile technology”, “wearable technology”, and “modern technology”, made the inclusion/exclusion criteria difficult to follow.

Line 27: extra question mark after 1

Line 44: “Second” there was not the first and third etc.

Line 46: missed citation

Line 47: the examples are not for treatment but diagnosis. It seems like a statement on diagnosis was missed.

Line 52-66: not all the technologies described are based on wearable and mobile technology, such as force plates and optical motion capture systems.

Line 98: This is not a systematic review paper

Line 98-111: those should be method sessions instead of introduction

Line 118-136: the automated filtering framework sounds exciting but very confusing here to connect them with the four steps stated in lines 128-136.

Line 318: should be reference 32 for the PICO

Line 622: Please elaborate on the limitations with a little more detail. What do you mean by available data sets?

Author Response

Please check the attached .pdf file.

Reviewer 3 Report

The article presents a systematic review of the research trends in technology usage for Parkinson’s Disease (PD) between the years 2008 (January) and 2019 (December). The authors summarize the highlighted works collected in four scientific databases. The reviewed works are organized according to applications, symptom analysis, and technologies applied in PD. The search methodology follows PRISMA and PICO statements to ensure the reproducibility of results. Also, the authors propose the use of an automated tool to pre-filter the papers of interest.

The major concern of this work is regarding the period in which (January 2008 and Dec 2019) the papers were collected. Although the great effort made by the authors is highlighted, the analysis of subsequent years is missing. An analysis that includes studies at least until December 2020 should be addressed, taking into account the advances that have been presented in recent years, especially in the area of wearable sensors and algorithmic area (i.e., machine learning).

General concept comments

The aims of this work can be of interest to the area of Parkinson's disease assessment and could complement the related literature and surveys. The research questions and main ideas are well supported by the related literature and include relevant references. The structure of the article is adequate, however, sections such as the methodology and the conclusion can be improved. Some, typo errors and mismatches in the number of the reviewed works in the databases should be reviewed in the figures, text, and supplementary material. Also, a few missing references were found. A thorough review of these points is suggested.

The methodology can be improved by performing a formal description of the inclusion and exclusion criteria for the paper’s selection under the PRISMA statements. Also, no description of the validity of the proposed tool (Phyton script) is identified or discussed.

In the results section, although the potential and trends in the use of mobile and wearable technologies in PD are clear, as commented by the authors using their findings and the information from previous works. A discussion about the possible challenges related to the use of these technologies could be addressed to identify possible gaps on which future work could focus.

The conclusions can be improved by a brief discussion of the main findings made in the previous sections and their implication for the current PD assessment. Also, in the conclusions section, a comment about the limitations on the availability of the data sets is stated, however, this is not described in previous sections (results), being an important consideration for the validation of systems intended for medical use.

In addition, the authors should address the following specific comments before the publication:

Specific comments:

Point 1: The introduction can be improved by avoiding describing part of the methodology (lines 100-104). Instead, this section should include a description of the objectives or motivation for conducting this study.

Point 2: A description of the validity of the proposed tool (Phyton script) should be presented and discussed.

Point 3: In the methodology section a description of the inclusion and exclusion criteria should be presented according to the PRISMA statement.

Point 4: In Figure 2, the data with the numbers of articles identified in the databases do not correspond to those described in section 4.2 (lines 308 to 309).

Point 5: Figures 7 and 8 could have the same magnitude on the color scale (maximum value) to facilitate comparison between figures.

Point 6: In the description of Figure 13, the word "heatmap" does not correspond to the figure. It seems that the description of figures 12 and 13 are crossed.

Point 7: In lines 583-584 the authors comment on the reasons that have led to the growth in the areas of wearable and biopotential devices However,  the statement about the progress in the area of sensors is not clear when referring to "low-processing" and "machine learning".

Point 8: Some missing references have been identified in the following lines:

  • line 27
  • line 46
  • line 63 VICON motion capturing system
  • line 641

Point 9: Minor typo and format errors have been found in the text, for example:

  • In lines 309 and 310 the sentence "..we removed the ones that were not in English.." could be changed to "..we removed the ones that were not written in English..".
  • Table 2 is out of margin.
  • Lines 126-127 are italicized without an apparent reason.

Author Response

Please check the attached .pdf file.

Reviewer 4 Report

This paper reviews the different technologies adopted for Parkinson’s disease assessment.

The paper is well written and easy to follow.

The systematic review methodology is well explained and produced good results.

The main issue of this work is that it does not summarizes the limitations and the drawbacks of the different approaches. The paper details different approaches, giving a lot of references to the existing literature, but fails to summarize the gaps that should be closed by future research.

Concerning the minor points, on lines 27 and 46 there are two “?” indicating a missing reference. On lines 183 and 197, please remove the colon after the end of the subtitles. On line 292, please change “introduced 2” with “introduced in Section 2”.

Author Response

Please check the attached .pdf file.

Round 2

Reviewer 2 Report

Thank you for addressing most of the previous comments. The paper reads better and flows better.

Here are some of my further comments:

Line 76: should be "presents". This sentence did not read right "Section 2 present the related for and background for our analysis." Please correct

Were Table 1 and Figure 1 from the other literature or this study? I do not recommend presenting results before the methodology. I still have difficulty following that session two was named background, but many of the contents were results.

In Figure 1, not all the topics are symptoms. Need be more accurate. In addition, can you revise the order of the figure to have motor signs and symptoms at the top, the mixed in the middle, and non-motor at the bottom?

It was stated that the automated filtering process script was shared but not included as an appendix or on Github.

Figure 4. What do you mean by new-technology publications?

Line 336-338: Figure 5 did not show that wearable and mobile technology are most frequently used for diagnosis and prognosis/monitoring.

I think it is essential to discuss the potential issues with the current approach. The numbers of publications are important, but the quality of the publications could be even more critical. At least it should be discussed how a lack of examination of the quality of the publications could impact the interpretation of the findings.

Reviewer 3 Report

REVIEWER COMMENTS:

 I am glad the authors followed my suggestions in almost all the parts. I also understand that some of my requests were hard to fulfill in the present document. The structure and the writing of the article are adequate. The introduction is concise but a few important references are missing.

 The article can be considered for publication after reviewing the following minor comments:

 Specific comments:

 Point 1:

The abstract should be in a structured format. For example, it lacks a clear statement about the methodology (PRISMA) employed to assess the review. The addition of this information could improve the consistency of the statement on line 8. “(…) Between January 2008 and December 2021, 31,718 articles were published in four databases: PubMed Central, Science Direct, IEEE Xplore, and MDPI.

 Point 2:

There are missing references in the Introduction ( lines 23-28) when the authors discuss motor and non-motor symptoms.

Point 3:

It is not clear how this work differs from others or complements the literature. There is a brief paragraph on this (at the end of the 2.1 Related Work) that does not sufficiently clarify the issue. A brief discussion of the main differences with other PD technology surveys should be included, or in the case, it should include a brief discussion of how this work complements the related literature.

Point 4:

 A minor typo error has been found in the Abstract: According to the context, there is an extra comma in line 7: (..) prognosis, and monitoring (…)

Reviewer 4 Report

Authors addressed all my comments.
